Habitat sharing and interspecies interactions in caves used by bats in the Republic of Congo

Labadie Morgane 1 morgane.labadie@gmail.com
Morand Serge 2 3
Bourgarel Mathieu 1 4
Niama Fabien Roch 5
Nguilili Guytrich Franel 6
Tobi N’Kaya 6
http://orcid.org/0000-0002-5213-3273 Caron Alexandre 1 7
http://orcid.org/0000-0002-2942-4531 De Nys Helene 1 4
1 CIRAD, UMR ASTRE , Montpellier , France
2 Kasetsart University, Faculty of Veterinary Technology , Bangkok , Thailand
3 Kasetsart University—Mahidol University, IRL HealthDEEP, CNRS , Bangkok , Thailand
4 UMR ASTRE, CIRAD , Harare , Zimbabwe
5 Laboratoire National de Santé Publique , Brazzaville , Republic of the Congo
6 Direction Générale de l’Élevage (Service vétérinaire), Ministère de l’Agriculture, de l’élevage et de la pêche , Brazzaville , Republic of the Congo
7 Universidade Eduardo Mondlane, Faculdade de Veterinaria , Maputo , Mozambique
Mavian Carla
Electronic publication date: 2025 Jan 9
Publication date: 2025
Volume: 13
Electronic Location ID: e18145
Received 2024 Jan 17; Accepted 2024 Aug 30
Copyright: © 2025 Labadie et al.
Copyright year: 2025
Copyright holder: Labadie et al.
License: This is an open access article distributed under the terms of the Creative Commons Attribution License, which permits unrestricted use, distribution, reproduction and adaptation in any medium and for any purpose provided that it is properly attributed. For attribution, the original author(s), title, publication source (PeerJ) and either DOI or URL of the article must be cited.
License URL: https://creativecommons.org/licenses/by/4.0/

Keywords: African bats, Human-wildlife interface, Bridge host, Camera-trap, Disease ecology

Funding: European Union Delegation FOOD/2016/379-660 EU and the WOAH for the implementation of the Action EBO-SURSY This study and publication were supported by the European Union Delegation under the Agreement [FOOD/2016/379-660], signed between the EU and the WOAH for the implementation of the Action EBO-SURSY. The funders had no role in study design, data collection and analysis, decision to publish, or preparation of the manuscript.

==============================
Bats play key roles in ecosystem functions and provide services to human populations. There is a need to protect bat populations and to mitigate the risks associated with pathogen spillover. Caves are key habitats for many bat species, which use them as roosting and breeding sites. Caves, bats and their guano also attract many other animals along trophic chains which might favor direct or indirect interspecies interactions. Two caves hosting colonies of insectivorous bats have been investigated in the Republic of Congo to characterize habitat sharing and interactions between bats, humans and animals. We set up a camera-trap monitoring protocol during 19 months at the entrance of and inside each cave. Our results demonstrated the richness and complexity of the species interactions around and within these caves. We identified and/or quantified mainly rodents, but also numerous categories of animals such as insects, birds, reptiles and carnivores using the caves. We investigated the temporal variation in the use of caves and the potential interactions between humans, wild animals and bat colonies. Our study contributes to the understanding of the interface and interactions, for the first time quantified, between cave-dwelling animal species, including humans. This knowledge is important to promote the conservation of cave ecosystems and better understand the ecology of infectious diseases.

Introduction

Bats are associated with emerging infectious diseases, yet they are also key species in ecosystems. They provide important ecological functions pollinating and dispersing the seeds of over 549 plant species, and regulating arthropod and insect pest populations, thereby limiting economic losses for farmers worldwide (Castillo-Figueroa, 2020; Ghanem & Voigt, 2012; Kunz et al., 2011; Ramírez-Fráncel et al., 2022). Bats are threatened by numerous human activities (destruction of their habitat, pesticides, hunting) (Frick, Kingston & Flanders, 2019; Furey & Racey, 2016; Tanalgo et al., 2023), but they are also victims of bad reputation due to their role in the transmission of zoonotic pathogens (Afelt et al., 2018; Banerjee et al., 2019; Calisher et al., 2006; López-Baucells, Rocha & Fernández-Llamazares, 2018; MacFarlane & Rocha, 2020). It is therefore important to understand bat ecology and their interactions with other wildlife and humans in order to better assess potential health risk to human populations and advance efforts in bat conservation.

Caves offer very specific light, humidity and temperature conditions that create ecological niches that benefit the many cave-dwelling species of animals, plants and micro-organisms (Gabriel & Northup, 2013; Kosznik-Kwaśnicka et al., 2022; Kovác, 2018; Tomczyk-Żak & Zielenkiewicz, 2016; Pacheco et al., 2020). The presence of caves in an ecosystem therefore influences local biodiversity through their contribution of specialized cave-dwelling species, but also attracts non-cavernicolous species through the feeding, foraging or hunting opportunities they provide.

Caves are key habitats for many bat species, which use them as resting and refuge places as well as breeding and parturition sites (Barros, Bernard & Ferreira, 2020; Kunz, 1982; Ormsbee, Kiser & Perlmeter, 2007; Meierhofer et al., 2022; Struebig et al., 2009). A single cave can host a high diversity of bat species. In addition, some bat species can gather in caves by hundreds or thousands (Tanalgo, Oliveira & Hughes, 2022). This habitat can also be important for many other animal species (e.g., insects, birds) that use them for different purposes (e.g., refuges, breeding, foraging). In a cave populated by bats, many direct and indirect interactions can occur between bats, animals (wild or domestic) and humans (Furey & Racey, 2016; McCracken, 1989). Wild animals take advantage of the presence of bat colonies as a food source, with numerous examples of predation by birds, small mammals or snakes, but also insects, e.g., centipedes (Mallick, Hossain & Raut, 2021; Mas, López-Baucells & Arrizabalaga, 2015; Molinari et al., 2005; Ridley, 1898; Scrimgeour, Beath & Swanney, 2012; Tanalgo et al., 2019). Animal pets, cats and dogs, can also be major predators of these bat colonies (Costa-Pinto, 2020; Merz et al., 2022; Oedin et al., 2021). In addition, predation by some of these animals, such as cats, could facilitate pathogen transmissions to human populations (Salinas-Ramos et al., 2021).

From time immemorial, humans have been using caves as places of refuge and/or worship (Bonsall & Tolan-Smith, 1997; Moyes, 2012; Straus, 1979). More recently, humans have been exploiting these habitats to extract minerals and guano, or as tourist attractions (Okonkwo, Afoma & Martha, 2017; Simons, 1998). Recent studies have suggested an increased risk of zoonotic transmission in highly disturbed environments characterized by a rise in the frequency and the intensity of wildlife-human interactions (Afelt et al., 2018; Allen et al., 2017; Becker et al., 2018; Johnson et al., 2020; Plowright et al., 2021; Rulli et al., 2017; Wilkinson et al., 2018). Indeed, events involving the transmission of pathogenic micro-organisms (fungi, viruses, bacteria or parasites) between cave bats and animals, including human, have been recognized (Kholik et al., 2019; Federici et al., 2022; Jurado et al., 2010; Karunarathna et al., 2023). Studying the animal-human interface to identify interactions and potential pathogen transmission routes is essential (Caron et al., 2021; de Garine-Wichatitsky et al., 2021). The emergence of zoonotic pathogens from bats has highlighted the need to study the interfaces between bats and humans, as well as the interactions that bats may have with other wild or domestic animals that could create potential transmission routes to humans. This is not only important for the field of disease ecology, but will also help to understand bat ecology, including the importance of bats in the trophic chain, and help to develop adequate bat conservation programs.

In the Republic of Congo, a camera trapping protocol was designed and set up in two caves hosting bat colonies to characterize the interfaces between cave bats, other animals and humans. Firstly, we described the communities exploiting the inside and outside of the cave using a species richness index. We hypothesized that different caves constitute different microhabitats and are therefore occupied or used differently. Inside the cave, cave-dwelling species should be the most represented, while at the cave entrance, outside the cave, species exploiting the cave interior, as well as other species not entering the cave, should be present. Secondly, the daily overlap in activity patterns between non-bat species and bats was characterized in order to identify times of day conducive to contact, competition or predation. We hypothesized that species with strong interactions will have overlapping activity. Finally, we investigated whether the non-bat species richness and diversity of taxa varied over time. We then hypothesized that seasonal variations may have an impact on species richness in both caves due to variations in food resources. More specifically, during the long rainy season, species diversity and richness may be positively impacted by the presence of large bat colonies compared to other seasons.

Materials and Methods

Ethic statements

All protocols were carried out with the approval of the Ministry of Forest Economy and Ethics Committee of the Ministry of Scientific Research and Technological Innovation in the Republic of Congo (N°212/MRSIT/IRSSA/CERSSA and N°687/MEF/CAB/DGEF-DFAP).

Study area

Our study took place between 2021 to 2023 in two different caves situated in the Niari and Bouenza Department, about 50 km away from each other, near the town Dolisie, in the South of the Republic of Congo (Fig. 1). This region is subject to four seasons: the short dry season (January and February), the short-rainy season (March to May), the long-dry season (June to August), and the long-rainy season (September to December) (Samba, Makanga & Mbayi, 1999). The landscape is mountainous and calcareous, favoring the presence of numerous caves and cavities. It is mainly composed of grassy savannah with patches of secondary forests and a patchwork of crops close to villages.

Figure 1 Map showing the location of the two caves in our study in the Republic of Congo.

Mont Belo cave is inside the Bouenza department and Boundou cave inside the Niari department. (Map Background Stadia Outdoors). Both caves located about 50 km from each other. Map credit: Leaflet | © OpenStreetMap contributors, CC BY SA © Stadia Maps, © OpenStreetMap Tiles © OpenStreetMap contributors.

Study sites

We selected the two caves on the basis of the following criteria: (1) annual presence of colonies of insectivorous and potentially frugivorous bats in the caves, (2) one site close to a rural settlement and the other less subject to human disturbance, (3) caves with few entrances/exits to facilitate surveillance by camera traps, and (4) sites that are easily accessible at any time of year, including the wet season.

The first cave, Mont Belo (MB), consists of several chambers with a main entrance. The cave is surrounded by a small patch of secondary forest, followed after by large variety of food crops (peanuts, cassava, tomatoes, etc.,) located at almost 5 km from the village (see Appendix S1 for a general diagram of the cave’s configuration). The second cave, named Boundou (BD), is a tunnel-shaped cave in the rock face, with a main entrance and a small exit at the end on the other side (see Appendix S2 for a general diagram of the cave’s configuration). This cave is also surrounded by a small secondary forest surrounded by a large grassy savannah. The nearest village is more than 5 km away, and human activity is much less than MB. Both caves are considered sacred by the local populations. BD cave, despite its sacred nature, is seldom visited by the local population, and to our knowledge there are no regular religious activities or pilgrimages.

BD cave hosts at least six genera of insectivorous bats (Rhinolophus sp., Coleura sp., Triaenops sp., Macronycteris sp., Miniopterus sp., Hipposideros sp.) and one genus of fruit bats (Roussettus sp.) (M Labadie et al., 2024, in preparation). Conversely, only four bat genera (fruit-eating: Rousettus sp. and insectivores: Rhinolophus sp., Miniopterus sp., Hipposideros sp.) were detected in the MB cave. Both caves are maternity colonies for insectivorous bat species (see Appendices S1 and S2 for location—presence of bat colonies in both caves). The abundance of these species varies according to the season and the species’ life cycle (direct observations and bat capture data, M. Labadie, 2024).

Camera trap data collection

The camera traps survey was conducted between September 2021 and March 2023. We used nine Moultrie M50 cameras (Moultrie, Birmingham, USA) deployed inside and outside the two caves (see Appendices S1 and S2 for the location of each camera trap in the two caves). We installed at least one camera trap in front of the main entrance of the cave (i.e., “outside”), another inside the cave in the main chamber, and additional cameras at the other entrances/exits which serve as passages for wild animals. In both caves, we did not cover the aerial exits (exit wells) which lead to the top of the caves, due to technical and safety constraints.

In total, four cameras traps were installed in MB cave (including one inside and three entrances—outside camera) and five cameras traps in BD cave (including one inside and four entrances—outside camera) over 19 months. The cameras were programmed to trigger automatically (high-sensitivity detector) by taking a picture followed immediately by a 30s video, and to re-trigger after a 5-min delay. This configuration enabled us to limit repeated triggering by the same individual and to visualize their behavior through a video. The 19 months study period was hampered by technical problems with certain camera traps (camera shutdown due to technical problems, presence of insects or dirt obstructing the camera lens, over exposure). During December 2021 and January 2022, the cameras inside the caves (one of the two cameras at MB and the single camera inside BD) experienced technical problems, resulting in unusable data (black images and videos). Other cameras suffered other technical problems and were replaced and re-started as soon as the problem was detected. During our field activities (approximately every 2 months, defined in this article as research activities), we restarted the cameras after our visit with new batteries and SD cards. For the other months when we were not present, a local guide was trained to come and restart the cameras at the beginning of the month (between the 2nd and 10th of the month) with new batteries and SD cards. In total, the camera traps at both sites operated for 159 months (19 months × 9 cameras = 171 minus 12 months of malfunction = 159 months).

Data preparation

All videos and images were integrated into Timelapse (Greenberg, 2023; Greenberg & Godin, 2015) to extract metadata and standardize the extraction of data from photos and videos. We used Megadetector (Beery, Morris & Yang, 2019; Fennell, Beirne & Burton, 2022), an artificial intelligence tool, to make an initial screening of visual material regarding the detection of animals. This software only works on photos, and indicates animal presence with a blue square, and human presence with a red square. Megadetector results can be integrated into Timelapse.

We defined one event of “detection” of a specific species or animal category as the presence of this species or animal category on a photo. One detection can be the presence of several individuals at the same time, for example a detection of two humans at the same time on an image.

A comparison of detections between Megadetector and manual analysis of the photos by ML was carried out on all photos recorded over the first 6 months of the study. Megadetector proved its effectiveness in detecting all presences (except insects and bats), despite a few detection errors (false detection on a landscape element, absence of detection for fast-moving bats). In total, Megadetector falsely detected the presence of animals and humans in only 2.3% of all photos. Hence, we decided to use Megadetector to reduce the time of treatment of the last 13 months. The videos linked to the pictures on which animals or humans were detected by Megadetector were viewed manually in order to verify the number of individuals and to describe the behavior of animals. When the number of individuals of a species or animal category varied between the photo and corresponding video, we retained the highest count of individuals for our analysis. As Megadetector can only be used on photos and not on videos, and is not parameterized to specifically identify bats and insects, we estimated that the detection of these two specific taxa was under-represented. Moreover, counting the number of insects and bats on video is feasible but complicated and time consuming. To avoid bias and to standardize our analysis, we decided to note the presence of insects and bats, but not to count the number of individuals.

Species identification

Species on pictures and videos were identified using zoological books (Kingdon et al., 2013; Sinclair & Ryan, 2003), or with the help of experts. If identification at species level was not possible due to poor image or video quality, identification was made at class or order level. As rodents were difficult to identify at species level, we classified them into three categories: small rodents (<60 cm without tail), large rodents (>60 cm without tail) and porcupines.

We set up two capture sessions in each cave in order to identify more precisely the rodent species present. Details are presented in Appendix S3. We also grouped birds according to their size: small (from 2 to 20 cm in length) or medium (from 21 to 60 cm in length), and raptors (owl and hawk species). To simplify the presentation of our results, we also categorized insects into two groups: (1) flying insects, which include midges, butterflies and bees, and (2) other crawling insects (cave beetles, crickets) and arachnids (spiders).

Our results are presented using common names for species or the categories mentioned above.

Categorizing human activities

During the course of our study, we consulted local communities, including landowners, village chiefs and local guides (non-standardized informal interviews) on several subjects of interest to our study, such as cave used and the wildlife species consumed by the local population. They informed us that MB cave was used for prayer rites, while no human activities were reported in BD cave. Local people acknowledged performing bat hunting (mainly large frugivorous bats) and guano harvesting but rarely in the two studied caves. Local communities have also pointed out that they consume other species of wildlife (bushmeat), including porcupines, rats and genets. Due to the high level of human activity at MB, it was decided with the local population to cover the cameras if necessary to avoid disturbing religious practices. In some cases, the presence of a human followed by the recording of a “black photo” for a certain period of time could indicate the presence of prayer activity. We viewed the 30s videos in order to identify and categorize the human behaviors detected in our study caves. We defined five distinct categories: (1) guano collection inside the caves, (2) hunting activity on bats present in the caves (we saw nets being laid inside the cave), (3) praying activity or religious rites, (4) our research activities such as changing cameras, capturing bats or collecting guano and (5) others, for all undefined activities (e.g., people who apparently just visited the cave), encompassing activities that could not be clearly categorized.

Categorizing animal species behaviors

The behavior of the animal was classified into five different categories: (1) moving, (2) foraging behavior (an animal appearing to look for something to eat in the cave), excluding the action of hunting, (3) interspecific interaction (i.e., hunting, feeding on another animal), (4) other behavior (i.e., defecating, grooming, sitting, observing) and (5) intraspecific interaction (i.e., chasing each other, fighting or mating). We did not categorize the behavior of insects, bats or unidentified animals or when the videos were of poor quality.

Statistics analysis

Selection of data

We identified detection events as independent if the detected species or animal category on simultaneous pictures or videos was different for each camera trap and if the detection interval between the previous and next detection was greater than or equal to 30 min (O’Brien, Kinnaird & Wibisono, 2003; Sollmann, 2018). All analyses were performed using the number of independent detections except for the graphs on animal and human behavior, where we used the exact number of individuals detected per independent event.

Analysis

All figures were produced using ggplot2 (Wickham, 2016), patchwork (Pedersen, 2023) and cowplot (Wilke, 2020) packages implemented in R software (R Core Team, 2023).

We presented the results as a function of the number of camera days, i.e., the number of detections divided by the sum of the number of days each camera was in operation per site, multiplied by one hundred (Rovero et al., 2013; Yasuda, 2004).

To quantify the temporal activities between the species or categories and the extent of temporal overlap, we used camtrapR package (Niedballa et al., 2016). The “activityOverlap” function was used to estimate the kernel density (non-parametric method) for study location (MB or BD cave, inside or outside), which calculates the probability density function of a detection distribution (Meredith & Ridout, 2014). The overlap coefficient (Dhat1) can vary between zero and one (no overlap = 0 and total overlap = 1) (Linkie & Ridout, 2011).

Three species richness indices with standard error were calculated for each study site and location (inside or outside the cave) using the vegan package (Oksanen et al., 2022). The first index, the Chao estimator, calculates an estimate of the total number of species, taking into account “rare” species. The second index, Jackknife, gives an overview of potential bias and variability by systematically removing samples, and the third index, bootstrap, will assess uncertainty by simulating sampling while providing a confidence interval and standard error (Chiu et al., 2014; O’Hara, 2005; Smith & van Belle, 1984).

We used a non-parametric Kruskal-Wallis test coupled with a Dunn’s test (post hoc test with Bonferroni correction) to test the effect of the four seasons and study location (MB or BD cave and inside or outside) on the number of detections for each taxonomic class (Dinno, 2015; Tomczak & Tomczak, 2014).

We also tested the existence of associations between the different species (with the most detections in each study site) using a Pearson’s product-moment correlation. Our data being tied, we were unable to use a non-parametric Spearman’s test.

Results

Dataset

In total, we collected 88,231 observations (including photos, videos and duplicate of one detection if different species) over a period of 19 months, of which 45,670 (51.8%) came from BD and 42,561 (48.2%) from MB. We detected the presence of animals or humans in 24% of observations (21,123 observations) with 12,001 (56.8%) detections from BD cave and 9,122 (43.2%) detections from MB cave. Of these 21,123 detections, 8,836 (41.8%) were bat detections, 8,218 (38.9%) were detections of other taxa, including humans and 4,069 (19.3%) were insect detections. Following selection of independent detections (>30 min interval), the dataset contained 11,581 detections, including 4,443 bats, 2,899 insects, 263 humans and 3,840 detections of others vertebrates (Appendix S4). The number of days of functional cameras varied between month, with an average of 16.4 days per month for camera traps at BD cave and 11.5 days for camera traps at MB cave (Fig. 2 and Appendix S5). Cameras inside the caves generally performed better than those outside (14.5 days vs. 12.2 days) (Appendix S5).

Figure 2 Number of days the camera traps were in operation during the study.

A total of nine cameras for all sites, with five for Boundou cave (including one inside the cave) and four for Mont Belo cave (including two inside the cave).

Presence of vertebrates and invertebrates in the two caves

In both caves, we mostly observed the presence of mammals (72.7% of all detections) including humans (2.3% of all mammalian detection), followed by insects (25% of all detections), birds and reptiles (Figs. 3A and 3B). Species richness and diversity were higher outside for both caves compared to inside (Tables 1 and 2). In the two sites, the accumulation curves for species richness showed that the number of species detected inside the caves reached an asymptote in contrast to the species detected outside the caves (Fig. 4). All taxonomic classes showed some variation of number of detections over the different seasons and the two study sites (Figs. 3A and 3B), but none was significant (p value > 0.05; Table 3). In both caves (inside and outside), species and diversity richness were highest during the long rainy season, followed by the long dry season and the two short seasons (wet and dry) (Table 2). The long rainy season was the season with the highest number of detections of any class. During the long and short rainy seasons, BD cave recorded a higher number of bird detections than MB cave (Figs. 3A and 3B). During the short rainy season, mammals and reptiles were in higher numbers in BD cave than in MB cave (Figs. 3A and 3B). Insects were detected in greater numbers in BD compared to MB cave during the short dry and short rainy seasons, but the opposite trend was observed for the other two seasons, long dry and long rainy seasons (Figs. 3A and 3B) in MB. During the short dry season, birds were not detected at MB (Figs. 3A and 3B). Reptiles were absent at BD during the long dry and short dry seasons (Figs. 3A and 3B).

Figure 3 Number of detections per camera day multiplied by 100 according to seasons and grouped by taxonomic class (bird, mammal, reptile and insect class) in (A) Mont Belo cave and (B) Boundou cave.

(A) Mont Belo cave, (B) Boundou cave. (N total = 11,581 detections with 8,422 mammals; 2,899 insects; 79 birds and 45 reptiles).

Table 1 Estimated species richness (Boostrap method with standard error, se) and species diversity richness (Shannon method with standard error, se) in relation to study location (Mont Belo and Boundou cave, inside and outside).

Study location	Total species	Boostrap ± se	Shannon ± se	
Mont Belo Outside	23	25.6 ± 1.6	4 ± 0.1	
Boundou Outside	17	19.3 ± 1.2	3.9 ± 0.1	
Mont Belo Inside	9	9.1 ± 0.3	3.8 ± 0.1	
Boundou Inside	9	9.2 ± 0.4	3.7 ± 0.1	

Table 2 Species richness index (Boostrap) and species diversity index (Shannon) in relation to season (Inside and Outside both caves) with standard error (se).

Cave location	Long rainy season	Short dry season	Short rainy season	Long dry
season	
	Species richness index: Boostrap ± se	
Mont Belo Inside	8 ± 0	9 ± 0	6 ± 0	9 ± 0	
Boundou Inside	9 ± 0	8 ± 0	8 ± 0	6 ± 0	
Mont Belo Outside	20 ± 0	11 ± 0	12 ± 0	16 ± 0	
Boundou Outside	13 ± 0	10 ± 0	12 ± 0	9 ± 0	
	Species diversity index: Shannon ± se	
Mont Belo Inside	1.3 ± 0.3	0.9 ± 0.4	0.8 ± 0.4	1.16 ± 0.4	
Boundou Inside	1.4 ± 0.3	1.4 ± 0.4	1.3 ± 0.4	1.24 ± 0.4	
Mont Belo Outside	1.7 ± 0.3	1.5 ± 0.3	1.7 ± 0.3	2.1 ± 0.2	
Boundou Outside	1.3 ± 0.3	1.1 ± 0.4	1.2 ± 0.3	1.5 ± 0.3	

Figure 4 Accumulation curves of species richness depending on study location.

(Mont Belo outside and inside, Boundou outside and outside) with confidence interval at 95%.

Table 3 Results of Kruskal-Wallis (non-parametric test) and Dunn’s test (post-hoc test) of classes identified between study location (Mont Belo and Boundou cave, inside and outside) and seasons (long rainy season (LRS), short rainy season (SRS), long dry season (LDS)).

Class	Study location	p value	LDS-LRS	LDS-SDS	LRS-SDS	LDS-SRS	LRS-SRS	SDS-SRS	
Insects	Mont Belo Inside	0.39	0.6	1	1	1	1	1	
Mont Belo Outside	0.39	1	1	1	0.6	1	1	
Boundou Inside	0.39	0.6	1	1	1	1	1	
Boundou Outside	0.39	0.6	1	1	1	1	1	
Mammals	Mont Belo Inside	0.39	0.6	1	1	1	1	1	
Mont Belo Outside	0.39	0.6	1	1	1	1	1	
Boundou Inside	0.39	0.6	1	1	1	1	1	
Boundou Outside	0.39	0.6	1	1	1	1	1	
Birds	Mont Belo Inside	–	–	–	–	–	–	–	
Mont Belo Outside	0.39	1	1	1	0.6	1	1	
Boundou Inside	–	–	–	–	–	–	–	
Boundou Outside	0.39	1	1	1	0.6	1	1	
Reptiles	Mont Belo Inside	0.39	1	1	1	1	0.6	1	
Mont Belo Outside	0.39	1	1	1	0.9	0.9	1	
Boundou Inside	0.39	0.6	1	1	1	1	1	
Boundou Outside	0.39	1	1	0.9	1	1	0.9	

More specifically, inside the MB cave, we mostly detected small rodents (62.2% of detection inside MB), followed by bats (18.3% of detection inside MB), other insects—arachnids (cave beetles, crickets, spiders) (10.5% of detection inside MB), large rodents (2.3% of detection inside MB) and flying insects (2.1% of detection inside MB) (Fig. 5). Inside BD cave, bats were most often detected (40% of detection inside BD), followed by other insects—arachnids (cave beetles, crickets, spiders) (35% of detection inside BD), genet (14.3% of detection inside BD) and flying insects (5.9% of detection inside BD) (Fig. 5). In BD cave, the genet, identified as the rusty-spotted genet (Genetta maculata) was one of the most frequently observed species both inside and outside the cave (Fig. 5). However, at MB cave, the genet, identified as the servaline genet (Genetta servalina) was not observed inside the cave, but only six times outside this cave (Fig 5).

Figure 5 Number of vertebrate species per camera days multiplied by 100 according to study sites (Boundou and Mont Belo cave) and position of camera (inside or outside).

Only the largest numbers of detections are shown in this figure (refer to the Appendix S6 to see all the detections).

Outside both caves, we identified the presence of species already detected inside the caves such as small rodents, genet, other insects—arachnids, large rodents, but also the presence of other species such as birds of different sizes (raptors), monkeys, pangolins and nile monitors (Fig. 5, Appendix S6). We also had a high level of detection of flying insects compared to inside the caves (14.4% of detection inside BD and 17.7% of detection inside MB) (Fig 5). The presence of humans was detected inside and outside both caves (Fig. 5).

In the MB cave, we observed a significant association between bats and flying insects (p < 0.001) as well as between small rodents and other insects—arachnids (p < 0.01) (Table 4). In the BD cave, we observed a significant association between bats and both types of insects (flying insects: p < 0.001 and other insects—arachnids: p < 0.001), and between both types of insects (p < 0.001) (Table 5).

Table 4 Association between detections of several animal categories (insects, bats, rodents) for Mont Belo cave.

Significant associations were tested using Pearson’s non parametric test (degree of freedom and significant P-value on the lower triangular table and correlation with 95 percent confidence interval on the upper triangular table).

Species	Other insects & arachnids	Flying insects	Bats	Small rodents	Giant pouched rat	
Other insects—arachnids	–	0.12 (−0.36–0.56)	0.30 (−0.17–0.67)	0.64 (0.26–0.85)	0.06 (−0.42–0.51)	
Flying insects	16, >0.05	–	0.95 (0.87–0.98)	−0.01 (−0.46–0.44)	−0.05 (−0.52–0.43)	
Bats	17, >0.05	17, <0.001***	–	0.22 (−0.25–0.61)	0.01 (−0.44–0.46)	
Small rodents	17, <0.01**	17, >0.05	17, >0.05	–	0.08 (−0.39–0.51)	
Giant pouched rat	16, >0.05	15, >0.05	17, >0.05	17, >0.05	–	
Note:

*** Highly significant association (p < 0.001).

** Very significant association (p < 0.01).

Table 5 Non parametric test of Pearson for Boundou cave depending on the species of interest, month and year of study.

Degree of freedom and significant P-value on the lower triangular table and correlation value with 95 percent confidence interval on the upper triangular table.

Species	Other insects & arachnids	Flying insects	Bats	Small rodents	Genets	
Other insects & arachnids	–	0.77 (0.48–0.90)	0.88 (0.72–0.95)	0.28 (−0.19–0.65)	0.33 (−0.14–0.68)	
Flying insects	17, <0.001***	–	0.92 (0.81–0.97)	0.001 (−0.45–0.45)	0.29 (−0.18–0.66)	
Bats	17, <0.001***	17, <0.001***	–	0.22 (−0.26–0.61)	0.43 (−0.02–0.74)	
Small rodents	17, >0.05	17, >0.05	17, >0.05	–	−0.01 (−0.46–0.44)	
Genets	17, >0.05	17, >0.05	17, <0.06	17, >0.05	–	

Appendix S7 provides a detailed list of the identified animals and the number of individuals per site. Some examples of images recorded for different species are also shown in Appendix S8.

Animal behavior inside the two caves—viewing video recordings

For 34% of the detected vertebrates (with the exclusion of bats, insects, humans and undentified animals; N = 7,742 detections), usable videos (i.e., enabling the action to be seen clearly, N = 500 detections) were analyzed to characterize their main behavior. For vertebrates counted in both caves (N = 1,146), we observed in ascending order: moving behavior (70.1% of all behavior observed), followed by foraging behavior (20.6%), interspecific interactions (5.7%), other behaviors (2.2%) and intraspecific interactions (1.3%) (Fig. 6).

Figure 6 Characterization of vertebrate behavior.

(Excluding humans, insects and bats) inside Mont Belo cave and Boundou cave.

Some pictures of Giant pouched rat show one or more individuals that appear to have swollen jowls (at least five different observations) (Appendix S9), which may suggest the use of the cave as a food storage site by this species. In the case of small rodents, we observed a few videos showing predatory behavior on the insects present in the cave.

The rusty-spotted genet regularly visited the inside of the BD cave and showed strong foraging and predation behavior inside the cave towards bats, rodents and insects. The servaline genet, on the other hand, was not recorded entering into the BD cave.

Concerning bird class, we observed the presence of the African wood owl (Strix woodfordii) at the entrance of the MB cave on ten occasions over a 19 months. Each time, videos showed that the individuals positioned themselves on a rocky promontory, close to the cave entrance, at around 18:00 to 19:00, a period of high bat flying activities at the entrance of the cave. In one of these videos, we were able to observe the hunting behavior of this owl as it dived towards the bats flying out from the MB cave. Details of the various interactions and potential contacts between the different species, including humans, in the two caves (inside and outside) are shown in Table 6.

Table 6 Summary of observed contacts, interactions, competitions and foraging behavior in the two caves (including inside and outside) between fauna and humans.

Species	Other insects & arachnids	Flying insects	Bats	Small rodents	Genets	Giant pouched rat	Human	Owl	
Other insects & arachnids	Interaction
competition	–	Contact potential	Foraging	Foraging	Contact potential	Contact potential	–	
Flying insects	Contact potential	Interaction
competition	Foraging
contact potential	Contact potential	–	Contact potential	Contact potential	–	
Bats	Contact potential	Foraging	Interaction
competition	Contact potential	Foraging	Contact potential	Contact potential	Foraging	
Small rodents	Foraging	–	Contact potential	Competition
interaction	Foraging	Contact potential interaction	Contact potential	–	
Genets	Foraging	Contact potential	Foraging	Foraging	Interaction competition	–	–	–	
Giant pouched rat	Contact potential	–	Contact potential	Interaction	–	Interaction
competition	–	–	
Human	Contact potential	Contact potential	Contact potential	Contact potential	–	–	Interaction
competition	–	
Owl	–	–	Foraging	–	–	–	–	–	

Human activities in the two caves

We observed more human activities in MB cave (n = 59.2% of the number of humans detected) than in BD cave (n = 40.8% of the number of humans detected). In MB cave, we detected in descending order: prayer activities (n = 38.8%), research activities (27%), other activities (i.e., undefined human activities) (22.8%), bat hunting (6.7%) and guano collection (4.7%) (Fig. 7A). The human presence at BD cave was observed for only two categories: our research activities (88.7%) followed by other undefined activities (11.4%) (Fig. 7B).

Figure 7 Description of human activities.

In (A) Mont Belo cave and (B) Boundou cave. Note the different scale on number of detections by camera days between (A) and (B).

Activity patterns of vertebrates of interest

We focused our analyses on the pattern of daily activity for vertebrates with the highest detection rates (>950 detections at both sites), with the exception of humans.

Genet exhibited a bimodal pattern of activities which peaked mainly at sunset (18:00) and a to a lesser extent at night until sunrise (between 03:00 and 06:00) (Fig. 8A). Bats and flying insect also showed a bimodal activity pattern, with a main peak of activities at sunset (around 18:00) and another peak before sunrise (between 04:00 and 06:00) (Figs. 8A and 8B). Small rodents showed a main peak of activities at sunset (around 18:00) and their activities remained fairly constant throughout the night, decreasing at sunrise (Figs. 8C and 8D). The highest activity overlap coefficient was observed between small rodents and other insects—arachnids (ground insects) (Dhat1 = 0.86), followed by genet and bats (Dhat1 = 0.84) and flying insects and bats (Dhat1 = 0.71) (Figs. 8A–8D). Details of activity at each survey location for the most frequently detected taxa are presented in Appendices S10 and S11.

Figure 8 Density of individuals in all study sites depending on time with (A) Activity overlap between genet and bats, (B) activity overlap between flying insects and bats, (C) activity overlap between other insects and small rodents and (D) activity overlap between bats and small rodents.

The daily pattern of human activity was observed to be unimodal, almost exclusively during the day (Fig. 9). Only prayer activities inside the cave showed activity with several peaks, including one at night (Figs. 9D and 9E). Table 7 summarizes the overlapping activities of the most frequently detected species (wildlife and humans).

Figure 9 Daily activity pattern of humans according to activities outside or inside the Mont Belo cave.

(A, B) Hunting activities outside and inside the cave, (C) Guano collection activities outside the cave, (D, E) Prayer activities outside and inside the cave, and (F, G) other activities outside and inside cave.

Table 7 Summary of the activities of the various species and humans in the two caves.

	Mont Belo	Boundou	
Time	Time	
Species	0–6 h	6–12 h	12–18 h	18–24 h	0–6 h	6–12 h	12–18 h	18–24 h	
Other insects & arachnids	X			X	X			X	
Flying insects	X		X	X	X		X	X	
Bats	X			X	X			X	
Small rodents	X	X		X					
Genets	X			X	X	X		X	
Giant pouched rat	X			X					
Human		X	X	X		X	X		
Owl				X					

Discussion

This study provides novel information on the use of the complex habitats offered by caves (Fenolio et al., 2006; Gnaspini, 2012; Simon, 2012) by animal species, including humans, on inter- species interactions (contacts and feeding activities) and seasonal variations of species richness and diversity.

It is, to our knowledge, the first comprehensive study of cave communities characterizing the interactions between bats and other animals, including humans. This study contributes to the understanding of the interface and interactions between animal species, including humans (Caron et al., 2021; de Garine-Wichatitsky et al., 2021). These results are important in the acquisition of knowledge on the ecology of infectious diseases, while improving our understanding of animal behaviour in these cave habitats. In addition, this type of study may help improving bat conservation programs (e.g., predicting the impact of environmental/biodiversity changes).

Species richness and diversity inside/outside cave

The study described the presence of three major taxonomic classes inside caves (mammals, insects and reptiles) and the addition of a fourth class, birds, outside caves. We detected greater species diversity and richness outside caves compared to inside, particularly during the long rainy season and the long dry season. The accumulation curves obtained from outdoor cameras showed that the 19 months of collection were not sufficient to detect all the species present in the surrounding environment. These results could be explained by two factors. First, each species has its own specific ecology (Beaver, 1979), and most species do not use caves regularly in their life cycle. Interestingly, we observed that certain animals, such as a primate, behaved fearfully towards these gaping holes in the rock (personal observation from a video of this study). Second, the absence of an asymptote for the species accumulation curve for outdoor cameras could be explained by the fact that our protocol focused on the passages of animals near the caves. However, as with all camera trap protocols, various parameters can have an impact on species detection, such as height and orientation of cameras, field conditions (vegetation, light, etc.,) and the difficulty of detecting certain species (e.g., snakes, tree-dwelling animals) (Trolliet et al., 2014; Burton et al., 2015).

This difference in richness, diversity and presence of taxa according to cave and season could be explained by the unique configuration of each cave (entrance shape, number of chambers, humidity, temperature, species present), external habitat (forest, savannah, presence of crops) and other factors (accessibility, proximity to inhabited areas, use of the cave by the local population) that create unique characteristics influencing the microhabitat in and around the cave (Gabriel & Northup, 2013; Gnaspini, 2012; Kosznik-Kwaśnicka et al., 2022). Seasonal variations have an impact on the availability of resources in a habitat, hence on the behaviour of individuals (movement, habitat use) (Aarts et al., 2013; Cumming & Bernard, 1997; Osman, 1978; Tonkin et al., 2017).

Despite the limitations of our protocol, which prevented us from accurately counting and identifying at species level the insects and bats present in the caves, during wet seasons, bats tend to be more numerous due to the synchronisation of the reproduction period with the period of high resource availability (Arlettaz et al., 2001; Nurul-Ain, Rosli & Kingston, 2017; Paksuz, Özkan & Postawa, 2008). Our results seem to show that their high abundance may favour the presence of predators (e.g., snakes feeding on young bats) or animals exploiting the guano during this period, such as insects (Di Russo, Claudio & Sbordoni, 1994; Jurado et al., 2010; Nitzu, 2021; Tlapaya-Romero, Santos-Moreno & Ibáñez-Bernal, 2021).

Our results showed an overlap between bat activity and flying insects inside the study caves (Fig. 7B). We also observed an association of bats with flying insects in the MB cave, and with both types of insects in the BD cave. These results may reflect and increased presence/activity of insects in the habitat at specific time such as dusk. Our result suggests a common period of their activities in both types of insects (flying and otherwise) and the microhabitat of cave (Gnaspini & Trajan, 2000; Royzenblat, Kulacic & Friedrich, 2023). Nonetheless, we cannot ignore the fact that the detection of certain mammals and insects is strongly related to our methodology (the positioning of the cameras and/or their sensitivity) (Palencia et al., 2021). In addition, data processing with Megadetector may also have had an impact on the detection of bats and insects, as it is not yet perfectly calibrated for this type of taxon (Choiński et al., 2023; Leorna & Brinkman, 2022).

Species identified and behavior

Four major groups of mammals are significantly represented: bats, rodents (small and large), genets and humans. As both caves are home to bat colonies, our camera traps detected a high level of bat presence and activity. The periods of flying activity of bats corresponded, as expected, with emergence at dusk and then return to the cave at dawn.

The high density of bats in a given area can favor the predatory behavior of certain species such as snakes, various carnivorous mammals and birds (Ridley, 1898; Tanalgo et al., 2019). For example, at the entrance of MB cave, we observed the presence of the African wood owl on ten occasions. This species was most probably predating on bats exiting the cave, as it has been observed previously flying to hunt (Kemp & Calburn, 1987).

The frequent detection of rodents (i.e., more than 20 detections in one night), especially inside MB cave, could be explained by the presence of food resources (i.e., insects) and the use of the cave by individuals as a refuge or for reproduction. Inside MB cave, we observed a high presence of several insect species that seemed to exploit the bat guano such as crickets and cave beetles. Several studies (Gnaspini, 2012; Sakoui et al., 2020) have shown the great diversity of insects that exploit bat guano litter. These insects could be a food resource for some of the small rodents. We made this assumption since several events of hunting behavior by rodents (N = 18) on insects were observed on video recordings. Moreover, we identified a species of small insectivorous rodent in the cave after setting a few rodent traps in the MB cave (pers. obs.; awaiting genetic confirmation of the species, see Appendix S3). In addition, the daily overlap activities of small rodents and other insects—arachnids also showed a significant relationship and strong association (Fig. 7C and Table 3). However, our photo-trap protocol allowed us to only measure the presence of insects inside and outside the caves through the frequency of detection. To obtain an accurate measure of insect abundance and the types of insects present according to the season, it would be necessary to set up an insect collection protocol.

We identified a Giant pouched rat (Cricetomys emini) which seems to use the MB cave as a food storage site. This result is in accordance with existing literature. This species is known to store its food in specific locations (Skinner & Chimimba, 2005; Tosso et al., 2018). The daily activity of Giant pouched rat was different from other rodents. It visited the cave regularly but showed irregular activity patterns (Appendix S11), which could also indicate food storage behavior in the cave.

Other frequently detected mammals were the rusty-spotted genet in BD cave (both inside and outside the cave) and the servaline genet outside of MB cave. Our results showed that rusty-spotted genets regularly visited the inside of the BD cave in search of food by predating on bats, rodents or insects, or by scavenging on bats (M Labadie et al., 2024 under submission). We also detected strong overlap of daily activity between genets and bats.

However, we did not detect any significant association between small rodents and genets. These results seem to validate the hypothesis of an opportunistic foraging behavior by genets in BD cave. The specific topography of the BD cave (Appendix S2) could also accentuate and favor bat hunting success.

Human activities

Our results show that human activities in the two caves varies greatly between MB and BD. In MB cave, the camera traps detected a high level of prayer activities over a period of 19 months, while BD cave is little or not used by the local population, as notified by the communities. In the MB cave, we also detected other human activities, including bat hunting and guano harvesting. Local people collect the bat guano as natural fertilizer for crops, a practice used in many countries around the world (Sakoui et al., 2020). Hunting activity was detected during two different months (March 2022 and January 2023). This irregular detection could be explained by a low hunting pressure due to the sacred nature of the cave and the almost exclusive presence of insectivorous bat species. One of the bat-hunting periods favoured by the hunters was January, which is a period of high fruit abundance for fruit-eating species, and the end of juvenile rearing for insectivorous bats. In our study area, close to the two caves, we were able to observe a quite significant hunting pressure on colonies of frugivorous cave bats.

Implication for disease ecology and for conservation

Disease transmission from reservoir hosts to humans depends on multiple factors including: (1) the distribution and density of the reservoir species, (2) the dynamics of the pathogen in the reservoir host, (3) the exposure of humans to the pathogen and (4) some internal factors of the person exposed to the pathogen (Ermonval & Morand, 2023; Plowright et al., 2017; Taylor, Latham & Woolhouse, 2001).

In Central Africa, caves are teeming with mammals and insects creating a specific microbiome both inside and outside the cave (Furey & Racey, 2016). The presence of bats and their guano provides attractive food resources for other species in the cave. The daily emergence of bats attracts opportunistic predators such as birds of prey, genets or reptiles outside the cave. Our results characterized inter-specific interactions and species involved in the trophic chain in these caves in the Republic of Congo. In the two caves, we described a trophic chain comprising the main bat-consuming species (inside and outside the cave) but also some other species that exploit the guano.

This work may help to investigate the potential mechanisms of micro-organisms transmission between different species; including some which might act as bridge hosts (Caron et al., 2015) between species known to carry a high diversity of micro-organisms (bats and rodents) and other at risk-species (humans, predators). At MB cave, several potential transmission routes of micro-organisms from bats to humans or other animals were identified, such as hunting and consumption of bats, exposure to aerosols through long presence inside the cave, the use of guano. This type of information is valuable in the field of health ecology, and can be used to further investigate the mechanisms and risks of inter- and intra-species transmission of zoonotic pathogens (Borremans et al., 2019; Cross et al., 2019; Ellwanger & Chies, 2021). Behavioral data can also help identify key informations for species conservation by: (i) studying population dynamics and dispersal, (ii) identifying priority habitats according to their use (feeding area, reproduction) and (iii) protecting habitats—areas of high connectivity (Berger-Tal et al., 2016; Caro, 2016; Greggor et al., 2016).

Limitation of our study

We faced several limitations due to field constraints. Due to the rapid movement of the animals, the quality of the videos and the environment (poor light and extreme humidity), it was difficult to identify the rodents down to species level (Burns et al., 2017). One way of solving this problem would be to use cameras inside rodent traps as suggested by Gracanin, Gracanin & Mikac (2019). We used size categorization for rodents. However, the size of individuals may vary according to the features of the camera field in which the individual was detected (near a rock, close to the camera or towards the back of the photo), its position in the photo or video (facing or from behind) and its behavior when the camera was triggered (e.g., running, sniffing or resting). Cameras placed inside caves also tended to operate for shorter periods due to the numbers of triggers caused by the movement of bats in the cave, as well as the presence of many insects. This problem may have limited the detection of some animals at certain periods. We also had to contend with a number of technical malfunctions due to the specificities of the cave environment (camera shutdowns, poor-quality photos and videos) which resulted in data loss. These various limitations have highlighted the improvements that could be made to this protocol to help increase data quality. This protocol could be improved by coupling it with additional protocols, such as insect trapping protocols and collecting faeces from certain animals (rodents, genets, owls) in order to analyze their diet and confirm some of our hypotheses.

Conclusions

This study is the first one to our knowledge that characterizes the interactions between bats, wild animals and humans in two caves in central Africa. Our results enabled us to provide a preliminary description of the multi-species communities sharing cave habitats and pave the way for further and optimized similar studies based on our experience.

A better understanding of these communities and interactions can help guide further research on pathogen transmission dynamics, as well as interventions to improve the conservation of bats and ecosystem services they provide. This work advocates for more research at the wild/domestic/human interface in order to cope with health hazards and promote coexistence between wildlife and humans.

Supplemental Information

Supplemental Information 1 Supplementary material.

We are very grateful to James Hogg, Olivier Hamerlynck and Zév Hamerlynck who identified the bird species and to Emmanuel Do Linh San for confirming the genet species. We would also like to thank the local authorities in Dolisie and the Congolese partners for their help during this project. Finally, we would like to thank the field team (F. Nguilili, C. Bazola, N. Nguimbi, R. Dimoukissi and R. Nguimbi) and the local communities who accepted the project.

Additional Information and Declarations

Competing Interests

Author Contributions

Ethics

Data Availability

The authors declare that they have no competing interests.

Morgane Labadie conceived and designed the experiments, performed the experiments, analyzed the data, prepared figures and/or tables, authored or reviewed drafts of the article, and approved the final draft.

Serge Morand conceived and designed the experiments, analyzed the data, authored or reviewed drafts of the article, and approved the final draft.

Mathieu Bourgarel conceived and designed the experiments, analyzed the data, authored or reviewed drafts of the article, and approved the final draft.

Fabien Roch Niama conceived and designed the experiments, authored or reviewed drafts of the article, and approved the final draft.

Guytrich Franel Nguilili performed the experiments, authored or reviewed drafts of the article, and approved the final draft.

N’Kaya Tobi conceived and designed the experiments, authored or reviewed drafts of the article, and approved the final draft.

Alexandre Caron conceived and designed the experiments, analyzed the data, authored or reviewed drafts of the article, and approved the final draft.

Helene De Nys conceived and designed the experiments, analyzed the data, authored or reviewed drafts of the article, and approved the final draft.

The following information was supplied relating to ethical approvals (i.e., approving body and any reference numbers):

All research protocol was carried out with the permission from the Ministry of Forest Economy and Ethics Committee of the Ministry of Scientific Research and Technological Innovation in the Republic of Congo (N°212/MRSIT/IRSSA/CERSSA and N°687/MEF/CAB/DGEF-DFAP).

The following information was supplied regarding data availability:

The data and code are available at CIRAD Dataverse: Labadie, Morgane, 2024, “Habitat sharing and interspecies interactions in caves used by bats in the Republic of Congo”, https://doi.org/10.18167/DVN1/TT3R02, CIRAD Dataverse, V1.

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
