# Peer review of "Habitat sharing and interspecies interactions in caves used by bats in the Republic of Congo"

_PeerJ, doi:10.7717/peerj.18145_

## Round 0.1 · original submission · Major Revisions

While the reviewers find the work a significant contribution to our understanding of potential transmission pathways by demonstrating potential interactions between bats, other animals and humans at a bat-human interface, there are some significant editorial issues with this manuscript. The authors will need to make some major changes in how this manuscript is written to improve the quality. The introduction, results and discussion are not linked. The discussion needs work to focus on the results of their own study and bring in other published work to support their findings. The conclusion needs to finish on a stronger note by clearly articulating what this study has shown and what the implications of these findings mean. It is lacking some key references and relies on summarizing a limited number of published works to set the scene. There are key references missing from this manuscript and there are several instances where a reference is not provided to support a statement. This manuscript needs to be improved by linking back to the literature.

Reviewer 2 has suggested that you cite specific references. You are welcome to add it/them if you believe they are relevant. However, you are not required to include these citations, and if you do not include them, this will not influence my decision.

Reviewer 1 ·

Basic reporting

The manuscript adheres to policies and for the most part is clear and conforms to professional standards. Some sentences are long and wordy and editing of sentences which contain multiple ideas would improve clarity. For several statements, references are missing and are needed to add to the quality of the manuscript. The article fits the standard sections of the journal submission structure.

Experimental design

The manuscript presents original research and fits within the aims and scope of PeerJ. The experimental design is generally sufficient to address the overall aims of the study. Additional details in the methods are needed to clarify some points. This is further outlined in the specific comments.

Validity of the findings

I felt there were several instances where the data was not adequately controlled. Several conclusions were not limited to supporting results. Several statements in the discussion might be better framed as author opinion because the results presented here do not support the conclusions they are trying to reach. While interesting and I largely agree with what has been written, I might argue that the research does not actually address transmission pathways and the authors are speculating based on the findings and/or broader published research.

Additional comments

Introduction
L41 “Bats are associated with emerging infectious diseases….”.
L50 Formatting issue. Remove _ _.
L50-52 Rephrase. Suggest something like “It is therefore important to understand the potential health risk to human populations, especially in human-bat interface habitats, to advance efforts in bat conservation.”.

Materials & Methods
L128 Add the word ‘of’- “near the town of Dolisie”.
L140 Change 5 to five.
L142-3 Suggest moving the sentence about human activity at Mont Belo cave to L138, where the cave is introduced. It would be clearer to present both environmental and human elements of each cave at once. The way it is written now, environmental elements are introduced for each cave, and then human elements are introduced. Suggest keeping the variables of each cave together.
L147, L153, L154 Remove s from ‘traps’ (camera trap, not camera traps).
L158 Remove ‘over’. “The nineteen-month study period was hampered by….”
L159 Remove s from ‘months’ (month, not months).
L161 Remove ‘…’.
L184 The results of the comparison between Megadetector and manual analysis should be presented to confirm that the use of Megadetector as the method of counting detections, is appropriate. Especially when L187 states that there are detection errors.
L195-6 Rephrase. Counting bats on videos as they emerge from roosts is feasible and a fairly standard method used by bat researchers. It is complicated and time consuming, but it is feasible.
L202 Add word “…identification was made to class or order level.”.
L203 Add word “… rodents were difficult to identify to species level”.
L204-5 Suggest just defining the categories used rather than including an example for one category. Also suggest keeping the way categories are presented, consistent for clarity. EG “small rodents (< 60 cm without tail), large rodents (> 60 cm without tail) and porcupines”.
L206-207 Suggest rephrasing categories again. EG “small (from 2 to 20 cm in length) or medium (from 21 to 60 cm in length), and raptors (owl and hawk species)”.
L213-216 This needs to be rephrased. These discussions are not interviews. They were non-standardised and likely not covered by human ethics approval. Suggest rephrasing to reflect the informal nature of these discussions. Perhaps ‘discussed’, ‘consulted’ or ‘liaised’. This engagement is important and it’s important to include it, but just reflect the informal way it was collected.
L217 Suggest replacing “local population” with ‘local community’.
L219 Add word “Presence of a human was followed by”.
L219-20 Add word “for a certain period of time which could indicate the…”.
L243 Bold heading.
L257 Add word “….the first index, the…”.
L259 Add word “….the second index, Jackknife…”.
L260 Rephrase “….the third index, bootstrap, assesses …”.

Results
L273 Change word “…detection of different species….”
L273 Remove s from ‘months’ (month, not months).
L290 I think the species richness accumulation curves are important to present in the results and not left to supplementary material. The graphs show you have reached an asymptote for cameras inside the caves but not outside which supports what you are presenting in the text and in Table 2. This gives confidence in the results you present for inside the caves, which is the focus of this manuscript. It is also an interesting result that species richness outside the cave has not reached an asymptote, indicating higher potential transmission pathways as different species may visit the entrance to the cave/s. This can be discussed in L397-399.
L306/309 Abbreviation of cave names (MB/BD) should be earlier in the manuscript and used throughout.
L313 and L315 Italicise scientific name.
L329-330 I thin the images should be moved to the Supplementary material. They are good to show as examples for interest, but don’t add to the analysis presented.
L339 Giant should be spelled with lower case g; giant pouched rat. Change throughout.
L340 Replace “underline” with “indicate”.
L341-342 Please provide the number of times this behaviour was observed (N=?).
L343-344 Rephrase. Suggest “….and showed strong foraging and predation behavior, and potential scavenging behavior….” To differentiate between strong behaviour and potential behavior.
L345. Remove reference. You are presenting your results here. Referring to other work to support your results happens in the discussion.
L346 Remove “into”.
L347 Italicise scientific name.
L348 Two-year period? The cameras were deployed for 19 months. Please provide number of times this predatory behaviour was observed (10- as referenced in L433). This is important data and the relative occurrence of this behaviour is valuable.
L355-356 Move sentence to L216, after mention of meeting with local community. While this sentence is a result, it is important to set the scene for differences between the caves. Introducing this information earlier will also provide more clarity to L217 where you explain why camera activity was modified at Mont Belo. Suggest starting the section on Human activities in the two caves with something like “As advised by the local community, Mont Belo had high human activity while the only human activity at Boundou cave was research activities associated with this study and….”. If there were only a few undefined activities, it would be beneficial to name them.
L369 Remove “a” after “and”.
L370 Flying insects?

Discussion
L383 Suggest removing “dwelling” as occupancy wasn’t the focus of this study. “Cave communities” is sufficient.
L385-386 Rephrase. Suggest “…this study contributes to advancing the understanding of the interface between wild and domestic animals….”.
L390 Remove “Even if the known events are rare,”.
L395 Rephrase. Suggest “… transmission routes, such as trophic chain, of…”. Change “illustrates” to “demonstrates”.
L397-399. I think these patterns need more discussion. This is more of a summary of results again, rather than a discussion about these patterns, and what they mean. L401-405 is a brief discussion on species richness results and an explanation that these results likely reflect microhabitat, but there is no real discussion on greater species richness outside both caves and the seasonal patterns.
L400, L407 and L417 Remove references to tables and figures through discussion.
L408. “denning” is not something insectivorous bats do so please reword to something more relevant.
L412-141 Suggest deleting “A specific protocol to count bats when they were flying out the cave was envisaged but it would have biased the presence and activities of species outside the caves.” As this is a separate discussion not developed in this manuscript.
L419-420 This result is supported by existing literature on the relationship between bat and insect activity. Suggest referring to some of that to support your findings and discussion.
L434 Suggest rephrasing to “…observed flying to hunt (Kemp…”. Statement can be further supported by referencing other published work on predation by owls on bats at caves.
L436 Add letter- times.
L439-440 Need to link to the references supplied. At the moment this sentence is discussing your result so some words to link your result and the relevant to the references provided is needed.
L441-44 This section is confusing. Do you mean hypothesis or theory? Is this a hypothesis you are testing (which is not mentioned in the introduction and should be if you are testing this) or is this a theory you are proposing in light of the evidence you have collected on rodents hunting insects? Are these records of rodents hunting insects from this study? This is the first time a rodent capture session has been mentioned. This section needs to be rewritten with clear messaging.
L447 Camera trap, not photo trap.
L451 Replaced upper case G with lower case g for giant and remove scientific name as that has been provided earlier. Change throughout discussion.
L455-456 Replace “… could also support the hypothesis of food storage…” with “… could also indicate food storage…”
L458-459 Remove scientific names as these have been provided earlier.
L462 Need to link result to reference. You need to present your result, explain it and support the finding with the reference.
L469 Tw-year period is mentioned again but the camera traps were only deployed for nineteen months. Please clarify.
L471 Here you are presenting results on human activities in Boundou cave but at L 355-356 you state that no human activities were reported in Boundou cave and at L363 you state other unidentified human activities were observed. These results are inconsistent. Please present clear results in the results section and then discuss them in the discussion, don’t bring in new results.
L483-489 I agree with this paragraph, but these statements need references, especially a reference to microbiomes and predation.
L487 Do you mean factors?
L494 Suggest “At Mont Belo cave, three are in contact,…”.
L503. Suggest rephrasing with “…. for example genet sp. which consume bats, or infection by aerosols….”.
L506 Replace “populations” with “communities”.
L502-506 I feel that the two bridge host examples need references to boost the validity of these points. I agree with them, but these aspects of transmission pathways were not part of your study and you do not have results to draw from to support these statements, so linking to references will validate these examples.
L506-509 This is confusing. Transmission has many variables, none of which have been explored here in detail and these points need references. I suggest making a general statement supported by references. Something like “However, transmission from an infectious agent to a reservoir to humans depends on multiple variables, including: (1) the distribution and density of the reservoir species, (2) the dynamics of the pathogen in the reservoir host, (3) the exposure of the human to the pathogen and (4) internal factors of the person in contact (add references)”.

Conclusions
L538-539 This sentence would be better placed after “….including humans.” And before your final sentence.

Figures
Figure 3- Suggest the legend identifying each class is presented in the order the graphs are presented (birds, mammals, reptiles, insects).

Tables
All tables are labelled Table 1 in the caption. This needs to be updated.

Supplementary material
Supplementary material should be presented in the order it is referenced. Currently Supp. 5 is the first material referenced, followed by 2,3 and 4. Suggest renumbering these into the correct order.

Reviewer 2 ·

Basic reporting

This paper provides an additional information on bat interactions within a cave system and attempts to discuss its implication to public health and disease transmission. While this holds an interesting promise, the current work did not really well showcased its findings. I believe this needs more rigorous work on framing of the narrative and consistency of the attempted topic (i.e., what do the authors really aim to convey at the end?). The introduction of the paper is too focused on disease transmission but the paper did not really delve into this. Similarly, the results of the study is new but limited that does not match the discussion of the paper. This needs a thorough work to avoid misleading your readers.

Experimental design

The experiment looks reproducible, the method section needs more clarity. The statistical test is not well presented and needs more work and clarity of variables.

Validity of the findings

Need to be clear on how the authors quantified and traced the behaviour of cave visitors. This aspect was presented but is not well elaborated.

Additional comments

Overall, the paper reports an interesting topic but needs more clarity and effort on the narrative and coherence of ideas. I also suggest to check recent articles on cave-dwelling bat conservation, for example:

Tanalgo, K. C., Oliveira, H. F., & Hughes, A. C. (2022). Mapping global conservation priorities and habitat vulnerabilities for cave-dwelling bats in a changing world. Science of The Total Environment, 843, 156909. https://doi.org/10.1016/j.scitotenv.2022.156909

Meierhofer, M. B., Johnson, J. S., Perez-Jimenez, J., Ito, F., Webela, P. W., Wiantoro, S., Bernard, E., Tanalgo, K. C., Hughes, A., Cardoso, P., Lilley, T., & Mammola, S. (2024). Effective conservation of subterranean-roosting bats. Conservation Biology, 38(1), e14157. https://doi.org/10.1111/cobi.14157

Annotated reviews are not available for download in order to protect the identity of reviewers who chose to remain anonymous.

·

Basic reporting

Authors can improve the manuscript by paying attention to small details of the text such as consistently placing spaces between values and SI units, italicizing species names, placing commas behind i.e., using commas in larger numbers etc. There are several opportunities throughout the mansucript to improve sentence structure to make them clearer. The citation of supplementary materials can be improved by referencing them in order of reading through the text.

These specific aims and hypotheses were not addressed by the manuscript or specifically discussed/answered in the text:
We hypothesize that different caves constitute different microhabitats and are therefore occupied or used differently.
Tthe overlap in activity patterns, on a daily basis, between non-bat species and bats was characterized in order to identify times of day conducive to contact, transmission, competition or predation. We hypothesize that species with strong interactions will have overlapping activity. Finally, using a non-parametric test, we verified whether the species richness of taxa varied over time. We hypothesize that seasonal variations may have an impact on species richness in both caves due to variations in food resources.

Experimental design

The research question is relevant and meaningful and addresses a large knowledge gap. It is unclear why certain decisions were made by the authors in the analysis of the data. I have highlighted these issues in an attached document.

Validity of the findings

No comment

Additional comments

I am unsure if it isa result of the online submission site, but all of the plots are stretched and the text in the figures is distorted. This should be fixed.

---

## Round 0.2 · accepted · Accept

Dear Authors,

both reviewers were pleased on how the comments were addressed and they believe that the manuscript is now ready for publication.

Reviewer 1 ·

Basic reporting

There are formatting issues relating to line spacing throughout, which should get picked up during editing but just be sure it does. For example, spacing issues at L130-131, L140-141, L150-152, L186, L190.

L153 Remove s from ‘traps’ (camera trap, not camera traps); “The camera trap surveys were conducted…”

L220 Formatting- suggest moving this up to L219.

L352 Giant should be spelled with lower case g; giant pouched rat. Change throughout. This species is mentioned first at L352 but the scientific name is provided at L483. Please remove the scientific name from L483 and provide after the common name provided at L352.

L407 ‘improving’ needs to be changed to ‘improve’.

L499 ‘varies’ needs to be changed to ‘varied’.

Experimental design

I think comments were answered well in the author response. I think this helped strengthen their results.

Validity of the findings

No comment- revisions have adequately addressed feedback.

Additional comments

The authors have sufficiently addressed my comments and suggested changes. This manuscript is acceptable for publication. I look forward to seeing the published article!

Reviewer 2 ·

Basic reporting

I believe the authors have addressed all my comments and suggestions well. I have no further comments.

Experimental design

I believe the authors have addressed all my comments and suggestions well. I have no further comments.

Validity of the findings

I believe the authors have addressed all my comments and suggestions well. I have no further comments.